# Characterizing the Cellular Response to Nitrogen-Doped Carbon Nanocups

**DOI:** 10.3390/nano9060887

**Published:** 2019-06-16

**Authors:** Amber S. Griffith, Thomas D. Zhang, Seth C. Burkert, Zelal Adiguzel, Ceyda Acilan, Alexander Star, William S. Saunders

**Affiliations:** 1Department of Biological Sciences, University of Pittsburgh, Pittsburgh, PA 15260, USA; ASG56@pitt.edu (A.S.G.); TDZ7@pitt.edu (T.D.Z.); 2Department of Chemistry, University of Pittsburgh, Pittsburgh, PA 15260, USA; scb55@pitt.edu (S.C.B.); astar@pitt.edu (A.S.); 3TUBITAK, Marmara Research Center, Genetic Engineering and Biotechnology Institute, 41470 Gebze/Kocaeli, Turkey; zelal.adiguzel@tubitak.gov.tr; 4School of Medicine, Koc University, 34450 Sarıyer, Turkey; cayhan@ku.edu.tr

**Keywords:** nanocups, nanotubes, biocompatibility

## Abstract

Carbon nanomaterials, specifically, carbon nanotubes (CNTs) have many potential applications in biology and medicine. Currently, this material has not reached its full potential for application due to the potential toxicity to mammalian cells, and the incomplete understanding of how CNTs interface with cells. The chemical composition and structural features of CNTs have been shown to directly affect their biological compatibility. The incorporation of nitrogen dopants to the graphitic lattice of CNTs results in a unique cup shaped morphology and minimal cytotoxicity in comparison to its undoped counterpart. In this study, we investigate how uniquely shaped nitrogen-doped carbon nanocups (NCNCs) interface with HeLa cells, a cervical cancer epithelial cultured cell line, and RPE-1 cells, an immortalized cultured epithelial cell line. We determined that NCNCs do not elicit a cytotoxic response in cells, and that they are uptaken via endocytosis. We have conjugated fluorescently tagged antibodies to NCNCs and shown that the protein-conjugated material is also capable of entering cells. This primes NCNCs to be a good candidate for subsequent protein modifications and applications in biological systems.

## 1. Introduction

Carbon nanomaterials have garnered great interest for their biological applications due to their ease of chemical functionalization [1], fine-tuned control over their length distributions [2], and ability for enzymatic biodegradation [3,4]. There have been emerging biomedical applications for these structures for use in bioimaging, biosensing, and drug delivery [5,6]. Despite intensive investigation into how changes in chemical and physical composition affect carbon nanomaterial biological compatibility, clearly defined correlations between these properties have been difficult to achieve. Therefore, complete cytotoxic profiles of materials designed for biological applications are essential in order to decrease undesired side effects.

Pristine carbon nanotubes (CNTs) have shown inherent toxicity to cells due to their high aspect ratios [7] and generation of reactive oxygen species [8,9]. Due to their hydrophobic nature, unmodified CNTs are prone to aggregation in aqueous environments [5,10]. The undesirable properties of pristine CNTs can be circumvented by functionalizing the surface of CNTs [11]. The addition of carboxy and/or amine functional groups to the surface of CNTs not only reduces the toxicity of the material but also bolsters its solubility in aqueous solutions [5,8,12]. Although these modifications are improvements to the use of CNTs, it has been noted that the synthesis process and the types of functional groups present on nanotubes may influence their interaction with cells [10,13,14,15]. This highlights the importance of determining how modified carbon nanomaterials interface with cells.

One potential avenue for mediating toxic behavior of CNTs is the incorporation of nitrogen dopants into the graphitic lattice of CNTs. Nitrogen-doped CNTs have shown dose dependent alterations of cell proliferation with 1.2 mg/mL doses showing increased cell proliferation while higher doses of 120 mg/mL decrease cellular proliferation [16]. Similarly, nitrogen-doped CNTs have been shown not to be toxic in murine models in comparison to undoped CNTs [17]. However, conflicting results also have shown that nitrogen-doped CNTs can exhibit changes in cell proliferation at µg/mL dosages [18]. The differences in reported behavior is most likely due to variations in synthesis conditions and biological characterization of unique nitrogen-doped materials is essential prior to their utilization. A model system for nitrogen doping is observed in nitrogen-doped carbon nanocups (NCNCs) which are carbon nanotubes with unique stacked-cup morphology [19]. NCNC are characterized by a unique chemical composition as the interior cavity is primarily hydrophobic and the exterior is heavily oxidized. Furthermore, the nitrogen dopants exist primarily around the open edge of the cup allowing for chemical decoration with gold nanoparticles to create a sealed capsule for drug delivery [19,20]. Specifically, for this material, functional studies have been done on mouse models and immune cells [19,20]. In these studies, the authors demonstrated that NCNCs are biocompatible in an organismal model, and that neutrophils are capable of degrading the material. Additionally, NCNCs can be loaded with therapeutic cargo and injected in vivo resulting in antitumor effects making them an attractive material for further biological applications [21]. Combining the drug delivery aspect of NCNCs with modification of the external surface with proteins could result in enhanced therapeutic effects. While undoped CNTs have been used to transfer peptides and proteins across cellular membranes, this effect has not been explored on NCNCs [22,23,24]. A full understanding of how NCNCs interact with cells, including its subcellular localization, remains to be determined. In this study we investigated the interactions of NCNCs with two cell lines; HeLa, a cervical cancer epithelial cultured cell line, and RPE-1, an immortalized epithelial cultured cell line. Our analysis shows that NCNCs are mainly uptaken via endocytosis and that a large proportion of them reside in vesicles in cells. We also determined that NCNCs are not bioactive, in terms of disrupting cellular activities like cell proliferation and respiration. We also demonstrated that IgG-conjugated NCNCs are successfully uptaken by cells.

## 2. Materials and Methods

### 2.1. Synthesis of NCNCs

NCNCs were synthesized by floating catalyst chemical vapor deposition (CVD) synthesis using a three-zone Lindberg/Blue furnace and a 1 inch diameter quartz tube as previously reported [25]. NCNCs were synthesized by injecting a liquid precursor consisting of 0.75 wt% ferrocene, 10 wt% acetonitrile, and 89.25 wt% xylenes at a flow rate of 1 mL/h with carrier gases of Ar at 127 sccm and H_2_ at 38 sccm. The first zone of the furnace was set at 250 °C for solvent evaporation and the second and third zones were set at 800 °C for growth of NCNCs for 60 min. NCNCs were removed by a quartz cross-plate with a double-sided razor blade and used for subsequent oxidation and separation. Ten mg of NCNCs were transferred to a 3:1 mixture of H_2_SO_4_/HNO_3_ and sonicated in a Branson 1510 bath sonicator at a power of 80 W for 8 h (CAUTION: the strong acid mixture is highly corrosive and requires careful handling and proper safety protection). The resulting oxidized NCNCs were collected and washed by filtration using a polytetrafluoroethylene (PTFE) membrane filter (200 nm pore size) and washed thoroughly with double distilled water. The resulting oxidized NCNCs were separated by probe tip sonication for 4 h (Fisher Scientific, Hampton, NH, USA, FB505) in an ice bath. NCNCs were functionalized with fluorescently labeled IgG through EDC/NHS functionalization. 1 mL of separated NCNC were suspended in 3.65 mL of 0.05 M MES buffer to a final concentration of 0.01 mg/mL. The separated NCNCs were stirred at room temperature with the addition of 100 µL of 100 mM EDC (1-Ethyl-3-(3-dimethylaminopropyl carbodiimide) and 250 µL of 100 mM sulfo-NHS. The reaction was stirred for 30 min before being collected by filtration. The NHS tagged NCNCs were resuspended in 1 M PBS buffer with brief sonication. The NCNCs were stirred again in the dark for 2 h with the addition of 10 µL of IgG added to the solution. The resulting IgG functionalized NCNCs were collected by filtration through a 200 nm PTFE membrane and resuspended in water for cell studies.

### 2.2. Synthesis of GFP IgG Conjugated NCNC

One mL of 0.1 mg/mL NCNC were diluted in 3.65 mL of MES buffer and stirred at room temperature. 100 µL of 100 mM EDC and 250 µL of 100 mM sulfo-NHS were added to the solution and allowed to mix for 30 min. The functionalized NCNCs were filtered through a 200 nm PTFE membrane and resuspended in 5 mL of 1 M PBS. The NCNC solution was stirred again at room temperature with 20 µg of fluorescently labeled IgG (150 kDa, Alexa Fluor 488) added to the solution. The solution was stirred in the dark for 2 h and filtered through a 200 nm PTFE membrane. The functionalized NCNCs were resuspended in 5 mL of nanopure water (18.2 MΩ·cm).

### 2.3. Cell Culture and Materials

HeLa cells were incubated with high glucose Dulbecco’s Modified Eagle’s Medium (DMEM) (Sigma Aldrich, Saint Louis, MO, USA) and 10% fetal bovine serum (Atlanta Biologicals, Flowery Branch, GA, USA) under 5% CO_2_. RPE-1 cells were incubated with DMEM/F-12 (Hyclone) and 10% fetal bovine serum (Atlanta Biologicals) under 5% CO_2_. Nocodazole and methyl-beta cyclodextrin were purchased from Sigma Aldrich. Cell incubation with NCNCs was conducted in culture medium containing 1% penicillin/streptomycin (Sigma Aldrich). Transmission electron microscopy (TEM) fixation materials were obtained from Electron Microscopy Sciences, Hatfield, PA, USA. Recombinant TNFα was acquired from Thermo Fisher Scientific, Hampton, NH, USA.

### 2.4. Cell Viability Assay

For water-soluble tetrazolium (WST-1) assay, cells were seeded at (3 × 10^3^) per well in triplicates on a 96 well plate and allowed to settle overnight. At each designated time point the media was removed and WST-1 (Sigma Aldrich) at a dilution of 1:10 was added and incubated for 2 h. Plates were then analyzed on an EL800 microplate reader (BioTek, Winooski, VT, USA) at 450 nm.

### 2.5. Crystal Violet Assay

Cells were grown on a 96 well microtiter plate and incubated with NCNC at various concentrations for 24 h. The media was removed and washed in autoclaved di water two times. Cells were then incubated with 0.5% crystal violet in methanol for 20 min on rocker of 20 oscillations per minute. Crystal violet solution was removed, and cells were washed with autoclaved di water four times and left to dry inverted overnight. Methanol was added and incubated covered for 20 min on rocker at 20 oscillations per minute. Absorbance were measured at 564 nm on an EL800 microplate reader.

### 2.6. TEM Preparation

HeLa cells were incubated with 5 μg/mL NCNCs at designated time points. For the co-incubation experiment, 20 nm gold nanoparticles in citrate buffer (Sigma Aldrich) were used. The gold was spun down and resuspended in medium for a final concentration of 5 nM. Cells were then trypsinized and fixed in 2% glutaraldehyde and 2% paraformaldehyde in phosphate buffer for 2 h. Cells were then washed three times for 10 min each in phosphate buffer, then incubated in 1% osmium tetroxide in phosphate buffer for 1 h. Cells were then washed five times 10 min each in phosphate buffer and then incubated in 50% ethanol in deionized water for 10 min. Cells were then incubated in 70% ethanol for 10 min and 100% ethanol twice for 15 min. Cells were then incubated in 100% propylene oxide for 15 min each and then allowed to incubate in a 50% mixture of embedding epoxy and propylene oxide with rotation overnight. The next day the solution was replaced with 100% embedding epoxy and allowed to incubate for several hours. Samples were then allowed to cure overnight in BEEM capsules (Electron Microscopy Sciences) in a 60 °C oven. Samples were then cut into 70 nm sections using an ultramicrotome and placed on copper electron microscopy grids. Samples were imaged on a Morgagni TEM microscope (Beaverton, OR, USA) with a camera.

### 2.7. Fluorescence Microscopy

Cells were grown on glass coverslips and treated with NCNCs at various time points. They were then fixed with 4% paraformaldehyde in 1X phosphate buffer solution (PBS) for 15 min. Cells were then washed three times in 1X PBS and then incubated with rhodamine phalloidin (Cytoskeleton) at a 1:1000 dilution for 15 min and washed twice with 1X PBS. Cells were then washed with deionized water for 5 min and then mounted on to glass slides with a mounting solution with DAPI included (Life Technologies, Carlsbad, CA, USA).

Mitotic arrest was achieved via incubation with 25 ng nocodazole for 4 h. Cytokinesis failure was induced by incubating cells with 35 μM psychosine for 24 h.

To determine the efficacy of endocytosis inhibition, HeLa cells were seeded on glass coverslips and treated with methyl-beta cyclodextrin for 6.5 h at a concentration of 5 mM. After 30 min pre-incubation, 4 uL of 25 mg/mL Texas red dextran per mL of media was added to the coverslips and incubated for 6 h. Cells were then fixed with 4% paraformaldehyde and washed with 1X PBS three times and mounted on glass slides using mounting solution with 4′,6-diamidino-2-phenolindole (DAPI). Slides were imaged on an Olympus microscope using a 100X oil emersion objective (Olympus, Tokyo, Japan).

### 2.8. Confocal Microscopy of NCNC-IgG Conjugates

Cells were grown on a glass coverslip and incubated with NCNC-IgG conjugates for 24 h. The media was removed and washed in 1X PBS five times. Cells were fixed with 4% paraformaldehyde in 1X PBS for 15 min. Cells were then washed three times in 1X PBS and then incubated with wheat germ agglutinin at 1:1000 dilution for 30 min and washed with 1X PBS two times. Cells were then mounted onto glass slides a mounting solution with DAPI included (Life Technologies). Slides were imaged on a Leica TCS SP5 Confocal and Multi-Photon Microscope; Leica Microsystems, Buffalo Grove, IL, USA.

### 2.9. Centrosome Staining

Cells were grown on glass coverslips and incubated with NCNCs for 24 h. The media was then removed and replaced with 1 μg/mL nocodazole for 4 h. Cells were then fixed with 100% methanol at −20 °C for 15 min. They were then washed 3 times with 1X PBS and incubated in blocking solution containing 1.5% bovine serum albumin (BSA) and 0.1% Tween in 1X PBS for 30 min. Coverslips were then incubated with primary mouse antibodies against γ-tubulin (Sigma Aldrich) at a dilution of 1:2000 in blocking buffer for 1 h. After washing 3 times in 1X PBS, coverslips were incubated with 488-Alexa fluorophore-conjugated secondary antibodies (Life Technologies) for 30 min. The cells were then washed twice, 10 min each in 1X PBS, once with deionized water for 5 min and mounted on glass slides using mounting solution with DAPI. Slides were imaged on Olympus microscope using a 100X oil emersion objective.

### 2.10. Western Blotting

Cells were treated with NCNCs at various time points and then lysed with RIPA lysis buffer. The lysate was then spun down at high speed for 10 min and the supernatant was collected and electrophoresed on a 4–12% polyacrylamide gel. Protein was transferred to polyvinylidene difluoride (PVDF) membrane and blocked in 5% milk in tris buffer solution with Tween (TBST) and then incubated overnight with rabbit primary antibodies against LC3 (Cell Signaling, Danvers, MA, USA). The membrane was then washed with 1X TBST and incubated with horseradish peroxidase (HRP) conjugated secondary antibodies (GE Life Sciences, Marlbourgh, MA, USA) for 1 h. The membrane was then washed with 1X TBST and incubated with luminol for 5 min and imaged on an Amersham western blot imager.

### 2.11. Autophagy Induction and Inhibition

RPE-1 cells were induced to undergo autophagy using a starvation medium containing 140 mM NaCl, 5 mM KCl, 1 mM CaCl_2_, 1 mM MgCl_2_, 1 g/L glucose, and 10 mM HEPES at a pH of 7.4 and sterile filtered. Cells were incubated in starvation media for 6 h and then incubated either in the absence or presence of bafilomycin for 2 h. Cells were then lysed for western blotting. Autophagy was inhibited using 500 nM wortmannin for 6 h.

### 2.12. Inflammatory Cytokine Detection

IL-6 secretion was assayed using an IL-6 ELISA kit from BD Scientific. HeLa cells were seeded 3 × 10^3^ cells per well in a 96 well plate and allowed to attach overnight. Cells were then treated with increasing concentrations of NCNC, or 1 ng/mL TNFα for a duration of 24 h. The supernatants were collected and spun down at 8 krpms for 5 min to remove particulate material. 100 μL of supernatant was used for analysis in the ELISA.

### 2.13. DHE Staining for ROS

HeLa cells were incubated with varying concentrations of NCNCs for 24 h. Cells treated with 500 μM H_2_O_2_ for 2 h was used as a positive control. Five μM dihydroethidium (DHE) (Sigma Aldrich) were added to cells 30 min before the end of the designated incubation time. Cells were washed with 1X PBS once and then fixed with 4% paraformaldehyde for 15 min. The cells were then washed three times with 1X PBS for 5 min each and then washed once with autoclaved water. The cells were mounted on glass slides using mounting solution with DAPI. Slides were imaged on a Zeiss Axio Scope A1 microscope, Oberkochen, Germany, using a 10X objective.

## 3. Results

### 3.1. Investigating NCNC Cytotoxicity

We first determined if NCNCs had detrimental effects on cells. We tested metabolic rates by using a water-soluble tetrazolium (WST-1) assay and cell proliferation rates using cell counts and treatment with crystal violet. Cells were exposed to increasing concentrations of NCNCs for 24 h, up to 10 μg/mL, at which time the culture media was opaque from NCNCs in suspension (Appendix A). At these varying concentrations, no toxicity was determined metabolically (Figure 1A) or proliferatively (Figure 1B) in cells treated with NCNCs when compared to untreated cells. We obtained similar results in RPE-1 cells, an immortalized epithelial cultured cell line (Appendix A). HeLa cells were then treated with NCNCs at 2 μg/mL over a span of 72 h. In comparison to control cells, cells treated with NCNCs did not show significant changes in metabolic activity (Figure 1C). Confirming the low toxicity of NCNC exposure, we saw no measurable change in the trend of cell proliferation over 72 h for cells treated with 2 μg/mL NCNCs when compared to untreated cells (Figure 1D). Thus, NCNCs showed no toxicity to cells in culture in a range of concentrations and times tested.

There have been studies that suggest that single-walled CNTs (SWCNTs) have the potential to interfere with mitotic machinery [26]. We hypothesized that if NCNCs were to interfere with cytokinesis, the frequency in multinucleated cells would increase. Alternatively, if NCNCs were preventing cells from entering mitosis, there would be a decrease in the population of cells undergoing mitosis. Lastly, we hypothesized that if cells were delayed in mitosis, there would be an increase in the mitotic index. We counted the mitotic populations of cells that were treated with 2 μg/mL NCNCs and saw no significant changes in the mitotic index of cells treated with NCNCs when compared to untreated cells (Figure 2A). As a positive control for increasing mitotic index, the microtubule disrupter nocodazole was used. This suggests that cells treated with NCNCs are capable of entering and exciting cellular division. We then compared the multinucleation frequency of HeLa cells treated with 2 μg/mL NCNCs against untreated cells. As a positive control, 35 μM of psychosine, an inducer for cytokinesis failure, was used. We determined that there was no significant difference in the frequency of multinucleation for cells that were treated with NCNCs and untreated cells (Figure 2B).

It has been stated in other studies that cells exposed to multi-walled carbon nanotubes (MWCNTs) elicited an inflammatory response [27]. HeLa cells were treated with increasing concentrations of NCNCs for a duration of 24 h to determine if NCNCs compel cells to secrete the inflammatory cytokine interleukin-6 (IL-6). The supernatants were then collected and assayed using an enzyme-linked immunosorbent assay (ELISA) for IL-6. Cells were also treated with tumor necrosis factor α (TNFα) as a positive control for induction of the inflammatory cytokine IL-6. None of the concentrations tested, up to 10 μg/mL NCNCs, showed a significant increase in IL-6 secretion compared to untreated cells (Figure 2C).

We also investigated whether cells produce reactive oxygen species (ROS) when incubated with NCNCs. Previous literature has shown that some variants of MWCNTs can produce ROS in various cell types [14,16]. HeLa cells were treated with varying concentrations of NCNCs up to 10 μg/mL for 24 h. As a positive control, HeLa cells were treated with 500 μM H_2_O_2_ for 2 h. The cells were incubated with dihydroethidium (DHE), a superoxide-sensitive dye, to determine the induction of ROS. Cells that were treated with NCNCs did not have a significant elevation in ROS when compared to untreated cells (Figure 3A,C,D,G). To ensure that cells were not compromised for inducing ROS, HeLa cells were co-incubated with 10 μg/mL NCNCs and 500 μM H_2_O_2_. Like the positive control, DHE staining was significantly higher in this treatment than untreated cells (Figure 3B,F,G one-way ANOVA *p* < 0.0001). This data suggests that NCNCs do not produce ROS when incubated with cells. Taken together, this our data suggests that NCNCs are not cytotoxic to cells.

### 3.2. NCNC Entry into Cells

We next examined the entry of NCNCs into cells. We found that NCNCs can be visualized using bright field microscopy (Appendix A). After treating cells with 2 μg/mL NCNCs for 24 h, we saw that the NCNCs often gathered around the nucleus within cells (Appendix A). The asymmetric clustering adjacent to the nucleus suggested NCNCs may be associating with the centrosome. NCNC clusters were located in the proximity of the centrosome marker γ-tubulin (Figure 4A) consistent with centrosomal association. We considered that if the clusters of NCNCs were associating with the centrosome they were most likely trafficked on microtubules, and disruption of microtubules would lead to their dispersal. Under normal conditions, the clusters of NCNCs were on average 5.6 microns away from the centrosomes. To disrupt the microtubules, we incubated cells with 1 μg/mL of the microtubule disruptor nocodazole for 4 h after treating HeLa cells with 2 μg/mL NCNCs for 24 h. After nocodazole treatment, the NCNC clusters were significantly further away from the centrosomes, with an average of 8.7 microns in distance (*t*-test *p*-value <0.0001, *n* ≥ 100 NCNC puncta per treatment group Figure 4B,C). This suggests that cells are trafficking NCNCs to centrosomes in a microtubule-dependent manner.

Transmission electron microscopy (TEM) analysis showed that a subset of NCNCs resided in the cytoplasm inside cells (Figure 5A), while a majority of them were found inside vesicles (Figure 5B). On average about 50–60% of the NCNCs were encapsulated in vesicles. This number remained constant over a duration of 48 h of 5 μg/mL NCNC exposure. Since we saw the majority of NCNCs in vesicles, we hypothesized that NCNCs may enter cells via endocytosis. Since it has been determined that colloidal gold is taken up via endocytosis [28,29], we hypothesized if NCNCs are entering via endocytosis, cells that are co-incubate cells with gold and NCNCs may have both materials residing within the same vesicles. We incubated HeLa cells with 5 nM of 20 nm colloidal gold and 5 μg/mL NCNCs for 6 h and imaged cell sections on the TEM. We indeed found frequent examples of vesicles that contained both gold and NCNCs (Figure 5C). To confirm entry via endocytosis, we incubated HeLa cells with NCNC in the presence of the endocytosis inhibitor methyl-beta cyclodextrin (MβCD) over a period of 6 h. Five mM MβCD was able to effectively reduce the uptake of fluorescently marked dextran in our control experiments (Appendix A). In untreated cells, there was on average 8.7 NCNCs per cell section. This number decreased to 1.4 NCNCs per cell section (*t*-test *p*-value < 0.0058 *n* = 2 experiments) when cells were incubated with NCNCs in the presence of MβCD (Figure 5D). A residual uptake of NCNCs in MβCD-treated cells and examples of NCNCs appearing to poke through the plasma membrane suggests that a subset of the NCNCs may enter the cell passively by diffusion (Appendix A). Interestingly, we also found that some vesicles containing NCNCs appeared distorted (Appendix A) and on occasion, NCNCs were piercing through the vesicle membrane (Appendix A). This may also account for NCNCs residing in the cytoplasm.

### 3.3. Autophagy Induction

Some of the NCNCs were in double-membraned autophagy-like vesicles (Figure 6A), and we wanted to ascertain whether NCNCs were inducing an autophagic response in cells, potentially due to stress from NCNC exposure. Autophagy is a dynamic process where a protein, LC-3, is incorporated into autophagosomes after lipidation. The protein is then degraded when autophagosomes fuse with the lysosomes for breakdown [30,31]. This process occurs at low levels in cells to maintain cellular homeostasis. RPE-1 cells, with lower levels of baseline autophagy as compared to HeLa cells, were exposed to 2 μg/mL NCNCs for 6 and 24 h either in the absence or presence of lysosomal inhibitor bafilomycin a, which inhibits the hydrolytic proteins within the lysosome and allows us to observe the rate of this process, which is also known as autophagic flux [32]. If autophagy is activated within cells, it is expected to have an enrichment of the lipidated form of LC-3. Alternatively, if autophagic flux is happening at a faster rate in NCNC treated cells, the inhibition via bafilomycin a would lead to a stronger enrichment of lipidated LC-3 compared to the negative control in inhibitor. NCNC treated RPE-1 cells were lysed and probed via western blotting for LC-3 levels. The treatments were compared to untreated and nutrient-starved cells as a negative and positive control for autophagy. Cells that were treated with NCNCs for 6 and 24 h appeared to have the same levels of lipidated LC-3 compared to the untreated control. Investigating autophagic flux in the presence of bafilomycin also shown no change in NCNC treated cells (Figure 6B). These data demonstrate that NCNCs do not induce an autophagic response and that NCNCs found in autophagic vesicles is due to the basal level of autophagy that occurs within cells.

### 3.4. Protein Conjugated NCNC Are Uptaken by the Cell

To use NCNC as a delivery scaffold for biological applications, we wanted to determine if protein attachment on NCNCs would perturb its entry into the cell. Fluorophore-conjugated secondary antibodies were conjugated to amino functionalities of NCNC via EDC NHS cross-linking reaction. Immunofluorescent microscopy observations confirmed that the fluorescent antibody was successfully conjugated to the NCNCs. HeLa cells were treated with IgG-NCNC conjugates for 24 h and stained with wheat germ agglutinin to delineate the cell boundaries. Confocal microscopy analysis showed IgG-NCNC conjugates within the cell membrane, which indicated that IgG-NCNC conjugates can enter the cell (Figure 7). The IgG-NCNC conjugates were localized around the vicinity of the nucleus. Confocal images also showed that the proteins remained attached to NCNCs based on brightfield and fluorescent colocalization. This indicates NCNCs are a good substrate for protein modification and subsequent application in a biological system.

## 4. Discussions

In order for carbon nanotubes to be used to their full potential, we must first gain a full understanding of how nanomaterials interface with cells. The main obstacle scientists face with using this material in biological systems is potential adverse reactions. Compounding this issue is the fact that the synthesis, length, and functional modifications on carbon nanotubes can influence how toxic they are [10,14]. This emphasizes that not all carbon nanotubes are expected to elicit the same cellular response, and that generalizations cannot be made from one nanomaterial to another [33].

In this study, we investigated how NCNCs, a variant of carbon nanotubes, interact with cells. We determined that NCNCs have minimal effects on cellular behavior, suggesting they may serve as a useful bioagent. It has been noted that some SWCNTs are capable of interfering with dividing cells and that both SWCNTs and MWCNTs can have cytotoxic effects [12,26,34]. We saw neither of these changes with NCNCs, as we demonstrated that NCNCs do not interfere with cellular metabolism or cell division through the cytotoxicity assays done in this study. We have also shown that NCNCs incubated with cells do not produce ROS. This conclusion is consistent with other findings that suggest nitrogen-doped carbon nanotubes are less bioactive than other varieties of CNTs [16,17,19].

We are able to visualize individual NCNCs via TEM to determine that a large proportion of the material resides within vesicles inside of cells. Light microscopy on cells treated with NCNCs has shown that clusters of material tend to accumulate in the vicinity of the centrosome, and that this phenomenon is microtubule-based. We also determined that NCNCs enter cells primarily via the endocytic pathway. Our findings are congruent with the current consensus that the uptake of CNTs is mainly energy-dependent endocytosis [20,35,36,37,38].

Interestingly, it has been stated that variations of CNTs (SWCNTs and MWCNTs) can induce an autophagic response which can potentially have detrimental effects on the cell [12,34]. Although we found a portion of vesicles containing NCNCs that takes the characteristics of autophagosomes, we determined that cells do not engage in a robust autophagic response when exposed to NCNCs. Previous studies on NCNCs have found that neutrophils are capable of degrading these nanoparticles [19], but future work needs to be done to determine whether this degradation is related to the vesiculation of NCNCs we saw in cells.

We have also shown that fluorescently tagged protein can be conjugated to NCNCs, and that this modification does not interfere with their entry into cells. This aids in the visualization of the material, which could potentially give rise to the use of this material for live cell imaging.

NCNCs appear to be a prime candidate for biological applications due to the absence of adverse effects on cells that are incubated with the material. However, the NCNCs were targeted to specific domains which may influence their usefulness. To advance the use of nanoparticles in the biomedical field, further development of our knowledge on how NCNCs can interact with subcellular compartments, and how this influences NCNC structure and associated materials, is needed. The entrapment of CNTs in vesicles that eventually enter the autophagic pathway can be useful for scientists working on diseases involved with lysosomal dysfunction. Some of the treatments for these diseases attempts to replace mis-functioning enzymes with functioning protein via intravenous infusion [39]. For example, therapeutic enzymes can be loaded into NCNCs and potentially distributed for uptake in cells systemically. This allows for a more concentrated dosage of enzymes into lysosomes compared to intravenous infusions. Drug delivery for treatment of these diseases may be designed to activate after NCNCs passage through the endocytic pathway and potentiate NCNCs as a useful tool in terms of drug delivery and disease treatment.

## Figures and Tables

**Figure 1 nanomaterials-09-00887-f001:**
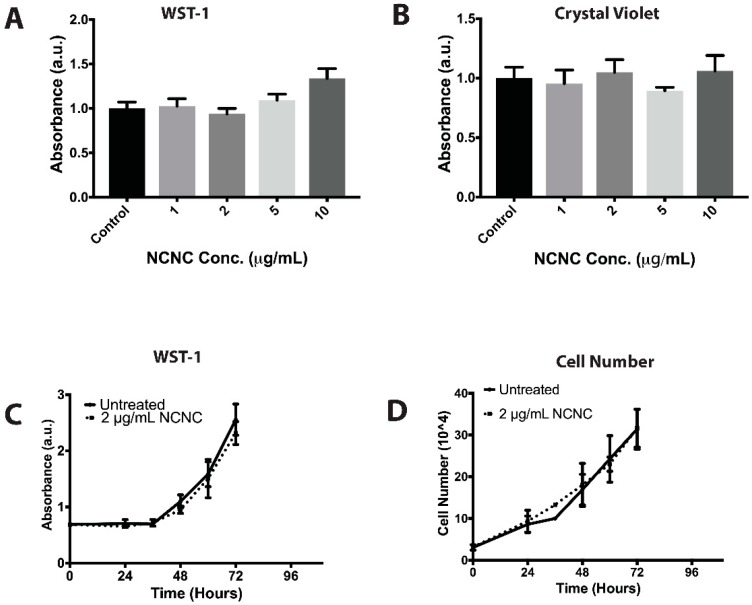
Metabolic and proliferation assays on HeLa cells treated with nitrogen-doped carbon nanocups (NCNCs). (**A**) HeLa cells were treated with increasing concentrations of NCNC for a duration of 24 h. After incubation, cells were assayed for metabolic activity using a WST-1 assay, *n* = 3 readings for each concentration. (**B**) HeLa cells were treated with increasing concentrations of NCNCs for a duration of 24 h. After incubation cells were assayed for viability using a crystal violet assay, *n* = 4 readings for each concentration. (**C**) HeLa cells were treated with 2 μg/mL NCNC over a span of 84 h. Cells were assayed with WST-1 at 12-h intervals, *n* = 3 readings for each time point. (**D**) Cell number was calculated on HeLa cells treated with 2 μg/mL NCNC over a span of 84 h using a hemocytometer, *n* = 2 counts for each time point.

**Figure 2 nanomaterials-09-00887-f002:**
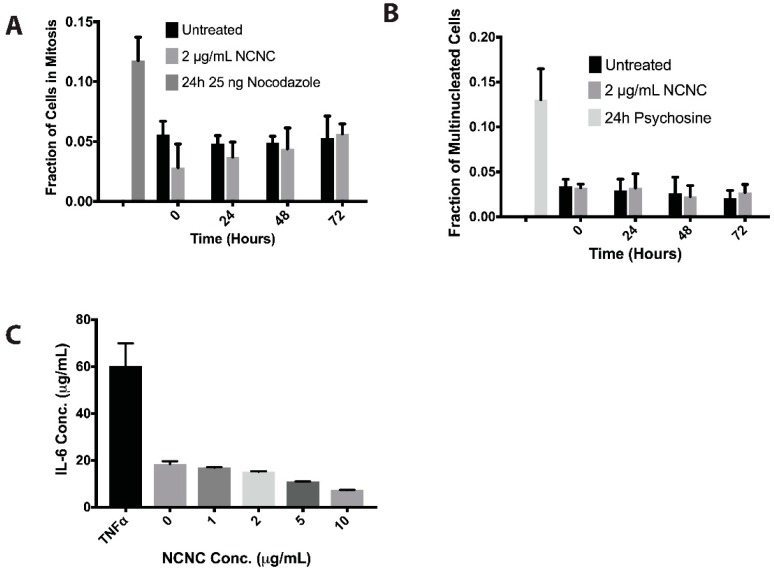
Mitotic and inflammatory response assays on HeLa cells treated with NCNCs. (**A**) HeLa cells were treated with 2 μg/mL NCNC over a span of 72 h. Mitotic indices were counted at 24-h intervals. HeLa cells were treated with 25 ng nocodazole for 24 h as a positive inducer for mitotic arrest, *n* = 3 separate experiments. (**B**) HeLa cells were treated with 2 μg/mL NCNCs over a span of 72 h. Multinucleation frequency was counted at 24-h intervals. HeLa cells were treated with 35 μM psychosine for 24 h as a positive induce of cytokinesis failure, *n* = 3 separate experiments. (**C**) HeLa cells were treated with increasing concentrations of NCNCs for 24 h. The supernatants were collected and assayed using an ELISA to probe for IL-6 secretion. 1 ng/mL of TNFα was used as a positive control for secretion of IL-6, *n* = 2 readings for each concentration.

**Figure 3 nanomaterials-09-00887-f003:**
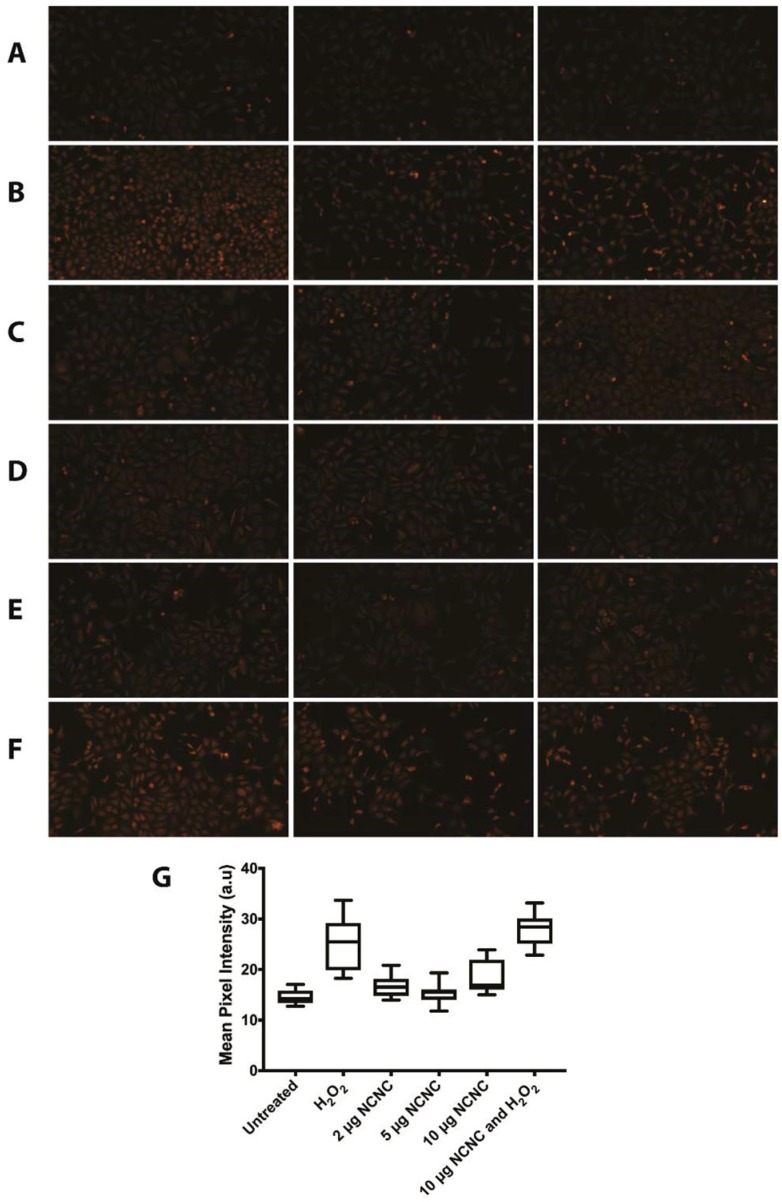
Dihydroethidium (DHE) staining for reactive oxygen species (ROS) in NCNC treated HeLa Cells. Pictured are representative images that are closest to the average pixel intensity for each treatment. (**A**) Untreated HeLa cells. (**B**) HeLa cells treated with 500 μM H_2_O_2_ for 2 h. (**C**) Hela cells treated with 2 μg/mL NCNC for 24 h. (**D**) HeLa cells treated with 5 μg/mL NCNC for 24 h. (**E**) HeLa cells treated with 10 μg/mL NCNC for 24 h. (**F**) HeLa cells treated with 10 μg/mL NCNC for 24 h and 500 μM H_2_O_2_ for 2 h. (**G**) Quantification of mean pixel intensity of images from treatments. Upper and lower bar represents maximum and minimum respectively. Central bar in box plot represents the mean. Each treatment has an *n* ≥ 7 images. For cells treated with 5 μg/mL NCNC, an outlier was omitted using the ROUT method for identifying outliers, reducing the sample set to seven images. The results remained the same if the outlier were to remain part of the data set.

**Figure 4 nanomaterials-09-00887-f004:**
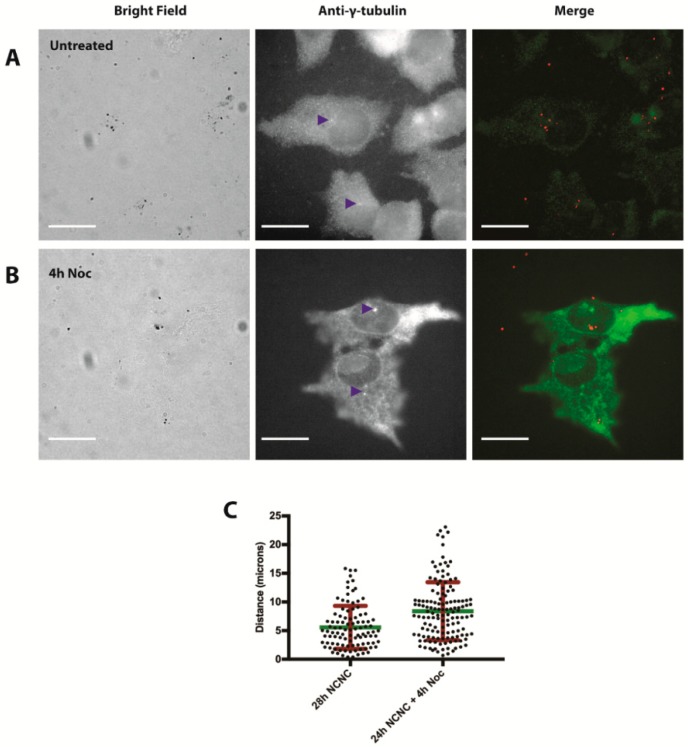
NCNC localization near centrosomes. (**A**) HeLa cells were incubated with 2 μg/mL NCNC for 24 h, they were then washed and incubated for 4 h with fresh medium and were fixed and stained using antibodies against γ-tubulin. Blue arrowheads indicate location of centrosomes. Red puncta in merge image represents the NCNCs imaged in bright field. (**B**) HeLa cells were incubated with 2 μg/mL NCNCs for 24 h, washed and incubated with the microtubule disruptor nocodazole for 4 h. The cells were then fixed and stained using antibodies against γ-tubulin. Blue arrowheads indicate the location of centrosomes. Red puncta in merge image represent the NCNCs imaged in bright field. (**C**) NCNC puncta distance was calculated using ImageJ software for cells in treatment groups from panels A and B. Green line indicates mean distance and red bars indicate standard deviation, *n* ≥ 100 puncta for each treatment group, *t*-test *p* value < 0.0001.

**Figure 5 nanomaterials-09-00887-f005:**
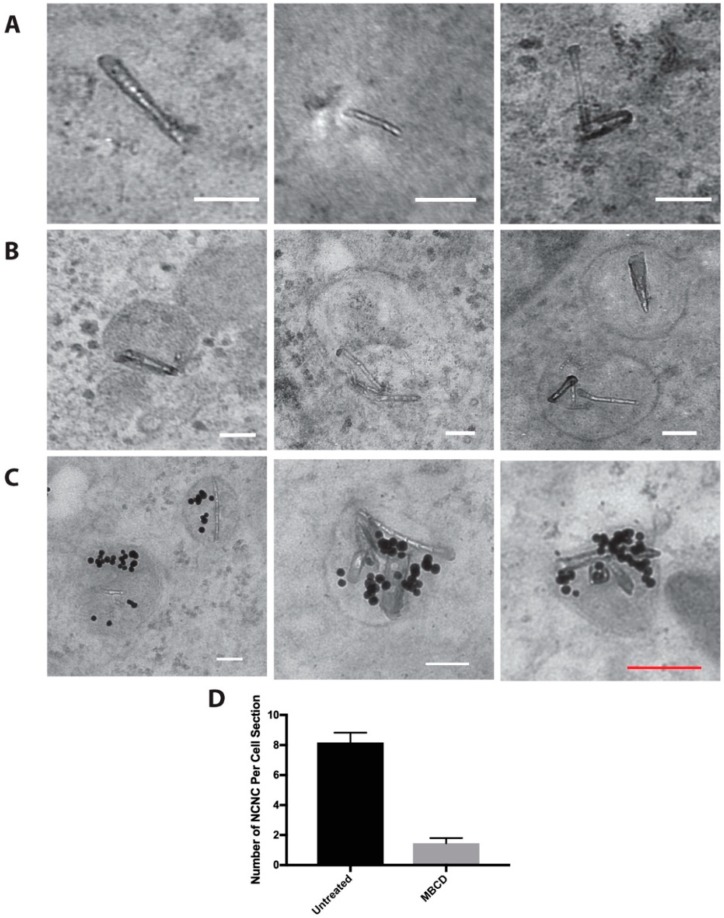
TEM images of NCNCs in cells. (**A**) Examples of NCNCs residing in the cytoplasm of HeLa Cells. Cells were treated with 5 μg/mL NCNCs for 6 h. (**B**) Examples of NCNCs residing in vesicles within HeLa Cells. Cells were treated with 5 μg/mL NCNCs for 6 h. (**C**) Examples of colloidal gold and NCNCs residing in the same vesicle. HeLa cells were treated with 5 μg/mL NCNCs and 5 nM 20 nm colloidal gold for 6 h. (**D**) HeLa cells were treated with 5 μg/mL NCNCs in the absence (untreated) or presence of endocytosis inhibitor MβCD for 6 h. NCNCs encountered per cell section was counted for each treatment, *n* = 2 separate experiments. *t*-test *p*-value = 0.0058. White scale bar indicates 100 nm, red scale bar indicates 200 nm.

**Figure 6 nanomaterials-09-00887-f006:**
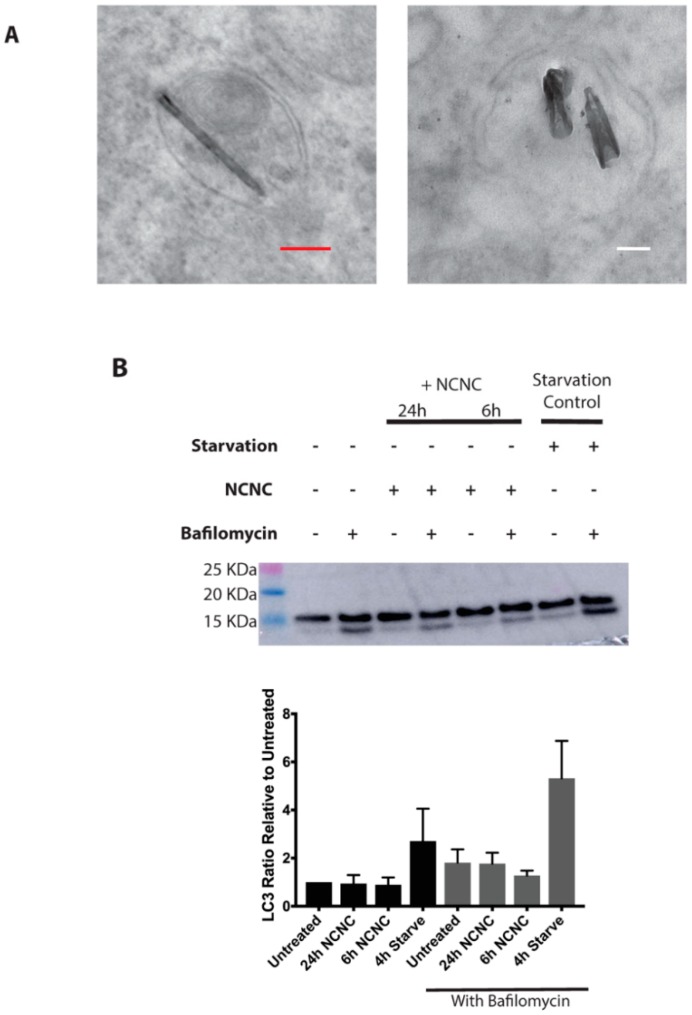
Autophagic response of cells exposed to NCNCs. (**A**) Examples of double membrane autophagic vesicles containing NCNCs in HeLa cells treated with 5 μg/mL NCNCs for 6 h (left) and 48 h (right). White scale bar indicates 100 nm, red scale bar indicates 200 nm. (**B**) Representative western blot proving for autophagy marker LC3. For NCNC treatments, HeLa cells were incubated with 2 μg/mL of NCNCs for either 6 h or 24 h. For starvation treatment, cells were nutrient starved for a duration of 4 h. Bafilomycin (100 nM) was added the last 2 h of treatment in specified conditions. Graph is a quantification of bottom band/top band ratios of four separate western blot experiments.

**Figure 7 nanomaterials-09-00887-f007:**
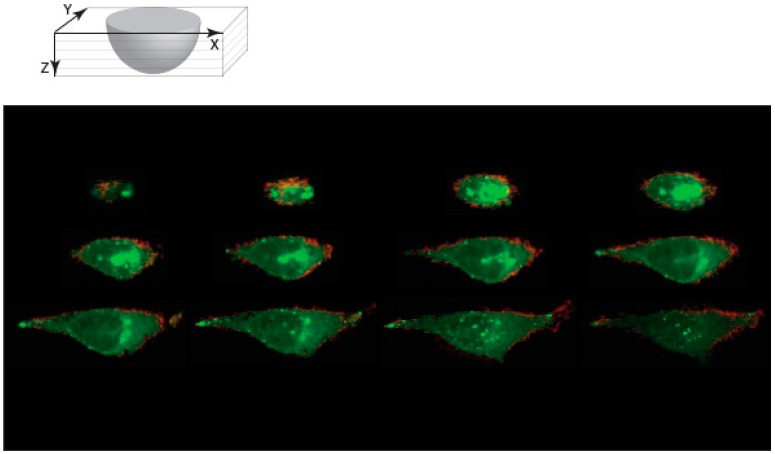
Confocal microscopy of NCNC-IgG conjugates entering cells. HeLa cells were incubated with fluorescently labeled antibodies (Alexa Fluor 488) conjugated to NCNC for 24 h and stained with wheat germ agglutinin. Z stack images were collected of a representative cell as shown. NCNC-IgG conjugates (green) are localized within the boundaries of the cell as shown by WGA (Red).

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
