# Peer review of "Characterizing the Cellular Response to Nitrogen-Doped Carbon Nanocups"

_nanomaterials, 2019, doi:10.3390/nano9060887_

Reviewer 1 Report

The manuscript “Characterizing the Cellular Response to Nitrogen-doped Carbon Nanocups” by Griffith et al. describes the preparation and in vitro characterization of nitrogen doped carbon nanocups. This is doubtlessly an interesting article which could be considered for publication. My minor comments are

1.       The authors have evaluated the cytotoxicity of NCNC over a narrow range of concentration (1-10 µg/mL). It is highly recommended to assess the cytotoxicity at higher concentration, such as at 100 µg/mL.

2.       Similarly, for mitotic and inflammatory response assay, the NCNC concentration was 2 µg/mL. Is there any specific reason to select 2 µg/mL? Please explain it.

3.       The “micro” sign is missing throughout the manuscript.

Author Response

Thanks to the reviewer for their speedy and helpful comments. I address them below.

Reviewer 1:

We disagree that the nanocups need to tested at concentrations above 10 micrograms/ml for the following reasons: At 10 micro/ml the cells are literally covered in nanoparticles.  We added a figure to the manuscript, Supp 1B, that shows this. This photo was taken after the cells were washed for immunofluoresence. So the incubated levels are even much higher. Note also in Sup 1A how opaque the solution is at this concentration. A previous publication with nanocups on mice used 20 micrograms for the entire mouse! Finally, the nanocups are precious and manufactured as needed by a collaborator (Star lab) so we can't just use a higher concentration without justifiable reason.

Ten micro/ml WAS actually used for the inflammatory response, but not for the mitotic index. Ten was meant to be an extremely high concentration, see above, not one to be used routinely. At that concentration, the great majority of NCs are wasted and never get inside the cell. So we didn't use it every time, specifically when we had sensitive assays that we could expect to show a response at lower levels.

The manuscript was revised and checked for formatting errors like the loss of micron.

Reviewer 2 Report

Report attached.

Author Response

Thanks to the reviewer for the timely and helpful comments. We added the first Chem. Rev. citation suggested and removed the Star acknowledgement. We added detail to the methodology of the NCNC conjugation. As indicated in the revision, the 488-Alexa was conjugated to the NCNCs. For protein concentration, we started out with 100 micrograms and resuspended the product in 5 mls of water as indicated in the revised manuscript. But we don't know the specific final concentration because we don't know the yields at each step. It is a small amount of novel hybrid (NCNC plus IG) material at the end that doesn't lend itself to common protein concentration assays. But, that being said, the main point is to show that functional conjugation of the NCNCs is possible. I would argue that doesn't require an exact knowledge of the final concentrations. In any case, it is beyond my technical skill to derive that information.

The honest answer to why pristine NCNCs instead of IgG conjugated were used is because we didn't know how to make that modification until after the initial experiments were completed. Also, I want the toxicity to be done with unmodified NCNCs. The IgG conjugation is just a proof of concept. Most of the time we will using other modifications for other biological purposes. I don't want the toxicity conclusions to apply only to IgG modified NCNCs. I would rather it apply as a baseline to unmodified NCNCs. Then we can decide if other modifications are likely to change that toxicity when the need arises.

Info on colloidal gold added as requested. RPE-1 were used as a (pseudo) diploid cell line to verify select conclusions in another cell line besides HeLa. Specifically for the apoptoic studies because the baseline in HeLa is too high. We used HeLa in most cases because of the better visualization in the microscope.
We made the changes suggested in the minor comments with these exceptions. We added column labels to Figure 4, but not 3 and 5. The reason is that columns in 3 and 5 are just additional examples of the same given conditions. Each column was treated the same. Using example #1 etc seemed clumsy.

We also left the scale bars as is. The reason is that each bar represents the same distance within a single figure. If we make them all the same length, then they would each be a different distance. I felt the original version was the least complicated. One scale bar is a different length and is in red to mark that.

Round  2

Reviewer 2 Report

The manuscript has been thoroughly revised and now it is suitable for publication in Nanomaterials.

Correct citation of reference [5] is: Advanced Healthcare Materials 2017, 6, 1700574.